# Insecticide Efficacy of Green Synthesis Silver Nanoparticles on *Diaphorina citri* Kuwayama (Hemiptera: Liviidae)

**DOI:** 10.3390/insects15070469

**Published:** 2024-06-23

**Authors:** Vidal Zavala-Zapata, Sonia N. Ramírez-Barrón, Maricarmen Sánchez-Borja, Luis A. Aguirre-Uribe, Juan Carlos Delgado-Ortiz, Sergio R. Sánchez-Peña, Juan Mayo-Hernández, Josué I. García-López, Jesus A. Vargas-Tovar, Agustín Hernández-Juárez

**Affiliations:** 1Departamento de Parasitología, Universidad Autónoma Agraria Antonio Narro, Calzada Antonio Narro 1923, Buenavista, Saltillo 25315, Mexico; vidal.convergens@gmail.com (V.Z.-Z.); luisaguirre49@hotmail.com (L.A.A.-U.); moe_788@hotmail.com (J.C.D.-O.); sanchezcheco@gmail.com (S.R.S.-P.); juan_013189@hotmail.com (J.M.-H.); 2Departamento de Ciencias Básicas, Universidad Autónoma Agraria Antonio Narro, Calzada Antonio Narro 1923, Buenavista, Saltillo 25315, Mexico; sonia.rmz.barron@gmail.com; 3Insectos Benéficos del Norte, Carretera Inter Ejidal, Camino Ejidal Libertad s/n, Ciudad Victoria 87260, Mexico; sanborjam@gmail.com; 4Investigador por México, Consejo Nacional de Humanidades, Ciencias y Tecnologías, Ciudad de México 03940, Mexico; 5Departamento de Fitomejoramiento, Universidad Autónoma Agraria Antonio Narro, Calzada Antonio Narro 1923, Buenavista, Saltillo 25315, Mexico; g.lopezj90@gmail.com; 6Tecnológico Nacional de México, Campus Instituto Tecnológico de Cd. Victoria, Boulevard Emilio Portes Gil 1301, Ciudad Victoria 87010, Mexico; armandoobiol@gmail.com

**Keywords:** AgNPs, pest control, nanoinsecticide, *Diaphorina citri*, asian citrus psyllid, green synthesis

## Abstract

**Simple Summary:**

Citrus production worldwide is threatened by the bacterium *Liberibacter* spp. Fagen and its vector *Diaphorina citri* Kuwayama. In the absence of an effective method against the bacteria, management plans are directed towards the vector. However, management strategies such as chemical control have presented difficulties in controlling the insect. Therefore, the application of new control techniques, such as nanoparticles by green synthesis, emerges as a promising control alternative. To address this issue, the objective of this study was to evaluate silver nanoparticles by green synthesis against *D. citri* nymphs in the laboratory and greenhouse. The results show that the nanoparticles achieved up to 100% mortality in the laboratory and 80% in the greenhouse. These results demonstrate the effectiveness of silver nanoparticles as a potential control method for the insect.

**Abstract:**

*Diaphorina citri* Kuwayama (Hemiptera: Liviidae) is a vector of *Liberibacter asiaticus* Jagoueix et al. and *Liberibacter americanus* Teixeira et al., causal agents of the critical yellow dragon disease or Huanglongbing (HLB), which affects citrus production worldwide. Recently, green synthetic nanoparticles have emerged as a potential alternative to control of agricultural insect pests. The insecticide effect of silver nanoparticles (AgNPs) on 2nd instar nymphs of *D. citri* under laboratory and greenhouse conditions was evaluated. Mortality was recorded 24, 48, and 72 h after application on *D. citri* nymphs under both laboratory and greenhouse conditions. The laboratory results showed that AgNPs caused 97.84 and 100% mortality at 32 and 64 ppm, respectively, 72 h after treatment. In the greenhouse, AgNPs caused 78.69 and 80.14% mortality using 64 and 128 ppm 72 h after application. This research is the first to evaluate the green synthesis AgNPs on *D. citri* and are a promising strategy to control the pest.

## 1. Introduction

Citrus are grown in tropical and temperate mediterranean regions of the world; these crops are highly consumed as fresh and processed products [1]. Citrus fruits are processed into juices, concentrates, canned products, as well as pulp, oils, and essences used in food and cosmetic industries [2]. Worldwide, production is estimated to be 183 million tons, with Asia being the leading grower with 51%, followed by America (29%), Africa (13%) and Europe (7%) [3]. The countries with the highest production are China, Brazil, India, Mexico, United States, and Spain. Mexico is the second largest exporter of lemons, with a production of 3,101,098.58 T and a value of MX$28,141,338.09 [4]; however, the crop with the highest production in the country is oranges, with a production of 4,854,373.26 T and a value of MX$16,142,238.20 [4]. With this production, the country ranks 4th in citrus production, 2nd in lemons and limes, as well as 5th in orange production [3].

*Diaphorina citri* Kuwayama (Hemiptera: Liviidae), known as the Asian citrus psyllid (ACP), is a pest of economic importance in citriculture [5]. The importance of the psyllid lies in the damage caused to citrus plants of the Rutaceae family (limes, oranges, grapefruit and tangerines) by carrying out its biological cycle on the plant [6,7]. Such damage includes feeding on young vegetative shoots, which, when found in large populations, causes the deformation of these shoots; in addition, the excreta of both adult and immature insects cause the development of fungi, which interfere with the photosynthesis of the plant [8,9].

In addition, the highest risk of *D. citri* is as vector of *Liberibacter asiaticus* Jagoueix et al. and *Liberibacter americanus* Teixeira et al. bacteria, which are a phloem-restricted bacterium and causal agents of Huanglongbing disease (HLB), or citrus yellow dragon [5]. HLB is considered one of the most devastating citrus diseases in the last two decades due to the losses caused in citrus production and eventually plant death [10,11]. Production losses are of up to 3.6 billion dollars annually in the case of the United States of America [12]; in Mexico, in the state of Tamaulipas, losses are estimated in 271,981.8 T with a value of 2,235,062.1 million pesos [13]. To date, there is no effective treatment to control the disease, so management strategies consist of removing infected plants and controlling the insect vector [14,15]. Chemical insecticides are mainly used; however, over the years, insect populations have developed resistance to various toxicological groups [16,17]; consequently, disproportionate applications of insecticides are used, causing a high environmental impact, due to the misapplications of these products [18]. The search for other control alternatives has been found, such as botanical insecticides and biological control [19,20,21,22]. In recent years, new technologies such as nanotechnology are making inroads into agriculture as a promising alternative to be implemented in integrated management programs [23]. Among these, nanoparticles (NPs), which have a variety of applications in agriculture, such as plant growth promoters, nutrient transporters, and potential insecticides, are being used [24,25]. Different types of NPs have been employed as nanoinsecticides, such as carbon nanotubes, metallic NPs as silver, zinc, copper, gold, titanium oxide, etc. [26,27].

Recently, green synthesis has emerged as one of the various methods of obtaining NPs from the use of microorganisms or plant extracts [28,29]. Such a technique offers a simple and environmentally friendly synthesis method compared to the chemical and physical methods employed for obtaining NPs [30,31]. Silver nanoparticles (AgNPs) are an efficient alternative in controlling economically important insects, i.e., Rehman et al. [32] synthesized AgNPs from *Camelina sativa* (L.) Crantz (Brassicaceae) and evaluated them against *Oryzaephilus surinamensis* (L.) (Coleoptera: Silvanidae) and *Sitophilus granarius* (L.) (Coleoptera: Curculionidae). On the other hand, Devi et al. [33] synthesized AgNPs from extracts of *Euphorbia hirta* L. (Euphorbiaceae), and these were tested against *Helicoverpa armigera* (Hubner) (Lepidoptera: Noctuidae), achieving an extension in the duration of the larval and pupal stage, as well as a reduction in male and female fecundity. In addition, AgNPs synthesized from extracts of *Citrus sinensis* (L.) Osbeck sweet orange peels have been tested against *Tribolium confusum* Jacquelin du Val (Coleoptera: Tenebrionidae incertae sedis), causing the high mortality of the insect, indicating that they possess great potential as a pest management alternative [34]. Therefore, this research objective was to evaluate the insecticidal potential of AgNPs synthesized from pecan nutshell extracts *Carya illinoinensis* K. Koch (Juglandaceae) on *D. citri* under laboratory and greenhouse conditions.

## 2. Materials and Methods

### 2.1. Experiment Location

The study was conducted at the Insectos Beneficos del Norte laboratory in Ciudad Victoria, Tamaulipas, Mexico.

### 2.2. Insect Rearing

Immature stages of *D. citri* were reared under greenhouse conditions (2 ± 42 °C). The insect colony was established in aluminum cages with 80 cm × 80 cm × 80 cm anti-aphid mesh, which contained 12 Mexican lemon plants *Citrus × aurantifolia* (Christm.) Swingle (Rutaceae) with young vegetative shoots. Plants were infested with 300 adult psyllids, which were allowed to oviposit for three days and removed. The colony was monitored until the nymph’s emergence and until they reached second instar, which were used for bioassays in the laboratory.

### 2.3. Synthesis and Characterization of Silver Nanoparticles (AgNPs)

AgNPs were synthesized at Ciencias Basicas Departament at Universidad Autonoma Agraria Antonio Narro in Saltillo, Coahuila, Mexico, by the green synthesis technique using pecan shells from *C. illinoinensis*, based on Neira-Vielma et al. [35]’s technique as follows: three grams of pecan shell powder dissolved in 300 mL of distilled water were placed in a spherical flask with three openings, and then connected to a refrigeration system and placed on a heating plate for four hours at 85 °C. The extract was then centrifuged, filtered, and refrigerated until use.

For the AgNPs synthesis process, a solution was prepared by dissolving 1.69 g of silver nitrate (AgNO_3_) in 985 mL of distilled water and 15 mL of the pecan shell extract. The solution was placed in a three-hole spherical flask to be taken to an iron for four hours at 95 °C and placed in amber glass vials for refrigerated storage.

The characterization of the AgNPs was carried out by Energy Dispersive X-ray Analysis (EDX) and Transmission Electron Microscopy (TEM) using a FEI-TITAN 80–300 kV microscope (Fisher Scientific, Hillsboro, OR, USA) at Centro de Investigación en Quimica Aplicada (CIQA) in Saltillo, Coahuila, Mexico.

### 2.4. Laboratory Evaluation of Insecticidal Effect on Diaphorina citri

Laboratory bioassays were conducted based on the Insecticide Resistance Action Committee (IRAC) methodology *002 (Psylla)* with some modifications. The experiment consisted of immersing vegetative shoots infested with second instar nymphs for five seconds in 50 mL of different concentrations of AgNPs (2, 4, 8, 16, 32, and 64 ppm), and distilled water was used as control. The immersed shoots were gently shaken to remove excess AgNPs and then placed in plastic trays with cotton discs moistened with distilled water for preservation. The experiment was established under a completely randomized design with 6 replicates per concentration and three experimental units per replicate, considering each vegetative shoot as an experimental unit. Mortality was evaluated at 24, 48, and 72 h after application, taking as the mortality criterion the response of the nymphs to the stimulus applied with a fine brush.

### 2.5. Evaluation of the Insecticidal Effect on Diaphorina citri in Greenhouse

Evaluation was carried out using 1-year-old Mexican lemon *C. aurantifolia* plants, obtained from a commercial nursery located in Ejido La Sanjuana, Güémez, Tamaulipas, Mexico., which were previously pruned to induce sprouting. Once shoots reach about 4 cm, they were infested with adult psyllids, which were placed in organza-type cloth bags; two pairs of adults were placed per shoot, left to oviposit for four days, and then removed. The eggs were monitored until the emerged nymphs reached second instar.

Tests consisted of direct application on the plants, using a 500 mL industrial hand sprayer, spraying plants homogeneously up to the runoff point. Based on laboratory results, four concentrations were used: 16, 32, 64, and 128 ppm, in addition to a control (distilled water). Six replicates per concentration were used and the mortality was taken 24, 48, and 72 h after treatment.

### 2.6. Statistical Analysis

The insect mortality record was subjected to a Probit analysis to determine LC_50_ and LC_95_ and fiducial limits with a 95% significance. Before the Probit analysis, mortality was corrected with Abbot’s [36] formula. Subsequently, to determine differences in mortality between treatments, the data were transformed (arcsine/square root) to be subjected to an ANOVA test under a completely randomized design for the laboratory, a randomized complete block design for the greenhouse treatments, and to a Tukey’s mean comparison (*p* < 0.05). Data analysis was performed in the SAS software OnDemand for Academics (online version). 

## 3. Results

### 3.1. Characteristics of AgNPs by Green Synthesis

EDX analysis shows that the composition profile of the AgNPs predominantly consists of silver (Figure 1). The absorption peak is close to 3 keV, based on surface plasmon resonance. On the other hand, the microphotographs and the histogram obtained by TEM show that the AgNPs have a hemispherical morphology with an approximate 50.2 nm diameter (Figure 2).

### 3.2. Mortality of 2nd Instar D. citri Nymphs in the Laboratory

AgNPs showed an insecticidal effect on *D. citri*. The results showed significant differences in mortality between concentrations, in which mortality increased as the concentration and time increased (Table 1). In the first evaluation at 24 h post-application, a maximum mortality of 45.83 to 55.21% was observed at concentrations of 32 and 64 ppm, with an LC_50_ of 40.89 ppm and an LC_95_ of 18,845 ppm, respectively. In the following evaluation (48 h), a considerable increase in mortality was observed, having from 66.70–90.66% depending on the concentration, with maximum mortalities at concentrations of 32 and 64 ppm, with an LC_50_ and LC_95_ of 0.23 and 189.25 ppm, respectively. The maximum mortality recorded was 97.84 and 100% at the highest concentrations 72 h post-application, with an LC_50_ and LC_95_ of 0.01 and 4.97 ppm, respectively (Table 1).

### 3.3. Mortality of 2nd Instar Nymphs of D. citri in Greenhouse

Evaluations under greenhouse conditions showed that AgNPs have a strong insecticidal effect on *D. citri*. Similar to the laboratory evaluations, mortality increased with increases in concentration (Table 2). Mortality at 24 h between concentrations showed significant differences (F = 34.89, g.l = 4, *p* < 0.0001); concentrations from 0 to 32 ppm, had the lowest mortality, from 2.08 to 10.6% at 24 h, whereas the highest mortality was recorded at concentrations of 64 and 128 ppm with 55.08 and 56.99%, with an LC50 and LC95 of 85.45 and 496.53 ppm, respectively. At 48 h there was an increase mortality per each concentration. Again, concentrations of 64 and 128 ppm showed the highest mortality with 64.19 and 70.14%, respectively, and LC_50_ and LC_95_ of 58.90 and 334.65 ppm. Finally, at 72 h post-application, the highest mortality recorded was 78.69 and 80.14% for the 64 and 128 ppm concentrations, respectively, and an LC_50_ and LC_95_ of 37.93 and 212.66 ppm (Table 2).

## 4. Discussion

Based on the results, it is shown that AgNPs by green synthesis have significant insecticidal activity on *D. citri* second instar nymphs under laboratory and greenhouse conditions, with this study being the first to evaluate AgNPs by green synthesis against *D. citri*. The insecticidal properties of AgNPs can be attributed to their morphological characteristics, dimensions, and high covering capacity, which favors their penetration into the insect body [37]. One of the advantages of employing AgNPs as a control agent is their low risk of developing resistance in prolonged use [38].

The mode of action of the insecticidal activity of silver nanoparticles can act in different ways, including increasing fat body protein synthesis, decreasing the level of detoxifying enzymes such as glutathione-s-transferase, neurotransmitter enzymes such as acetylcholinesterase and an increase in the production of reactive oxygen species (ROS) [27,39,40].

Information related to evaluation of AgNPs on *D. citri* is null under greenhouse conditions; however, there are other evaluations on insects of sucking and feeding habits. For example, when evaluating AgNPs synthesized from *Solanum lycopersicum* L. (Solanaceae) on the rose aphid *Macrosiphum rosae* L. (Hemiptera: Aphididae) mortalities of 50–100% were found at concentrations of 500 ppm [41], similar to reports in this research; however, at a considerably lower concentration (64 ppm). These authors mention that the effectiveness of AgNPs is attributed to the accumulation of the nanomaterials, which affect insect growth depending on the physicochemical properties, and the ions released by the interaction of AgNPs [41]. Similar cases are reported with the evaluation of AgNPs synthesized from extracts of *Azadirachta indica* A. Juss. (Meliaceae), which caused mortalities of 91.95 and 84.40% in the fourth and fifth nymphal stages of *Oxycarenus hyalinipennis* Costa (Hemiptera: Lygaeidae) at a concentration of 1250 ppm and 72 h post-application [42]. AgNPs synthesized from *A. indica* have also demonstrated insecticidal activity on other Hemiptera, such as whitefly *Bemicia tabaci* Genn. (Hemiptera: Aleyrodidae), on which they recorded mortalities of 80–82.22% at 72 h at a concentration of 1200 ppm over third instar nymphs [43]. These results are similar to those obtained in this research under laboratory conditions; however, at a considerably lower concentration (64 ppm).

AgNPs have been tested on other insects of different habits, such as *Spodoptera litura* (F.) and *Helicoverpa armigera* (Hubner) (both, Lepidoptera: Noctuidae), which report up to 78.49% mortality in larvae, causing histological changes in the insects’ gut [44]. The damage in insects can be attributed to the nanoparticle application method, for example, AgNPs supplied by the route of ingestion cause damage at the intestinal level, whereas contact in *T. confusum* caused abrasive damage to the insect cuticle by the loading of the nanoparticle, triggering dehydration in the insect [34]. On the other hand, it has been observed that the activity of certain enzymes, such as glutathione-s-transferase, catalases and dismutases, increases as a function of the dose; such is the case of *Sitobion avenae* Fabricius (Hemiptera: Aphididae), whereby, when increasing the dose of AgNPs, the enzymatic activity of glutathione-s-transferase and catalase was found to be dependent on the dose supplied, causing a mortality of 93.33% at 600 ppm [45].

Metallic nanoparticles possess important characteristics which make them useful in many fields; among these characteristics is the large surface area/volume ratio which allows a smaller amount of ingredient to be used per surface area, as compared to a bulk material [46,47]. Within metallic nanoparticles, we can find gold, copper, palladium, platinum, and silver, the latter being the most used for green synthesis due to its easy reduction through plant extracts [48]. The chemical synthesis methods of nanoparticles are among the most widely used, however, they usually use highly concentrated stabilizing agents in addition to demanding a high amount of energy [29]. In comparison, green synthesis is performed in a single step, which decreases its energy demand for synthesis; in addition, plant extracts are more environmentally friendly [49].

Recently, the implementation of the biological synthesis or green synthesis of nanoparticles has increased due to its multiple advantages, such as lower energy consumption, the replacement of complex equipment for their synthesis by simpler ones, as well as a lower release of toxic substances for the environment and human health [50]. This type of synthesis can employ organisms such as fungi or bacteria, as well as plant extracts as reducing stabilizing agents [51]. Plant extracts contain different compounds, such as alkaloids, terpenoids, phenols, tannins, and proteins, which possess a reducing and metal ion stabilizing capacity [52].

The main advantage of using plant extracts for the synthesis of nanoparticles is the availability of plants, in addition to the fact that they have low, or no toxicity as compared to other methods [29]. On the other hand, due to the wide range of secondary metabolites in the extracts, they help in the reduction of the materials in a faster way [46]. Finally, flavonoids, steroids, and carbohydrates provide coating activity and stability to the metal nanoparticles [53].

Although silver is not an essential element for citrus development, it can cause beneficial effects, such as promotion in vegetative bud development, an increase in total chlorophyll, tolerance to biotic stress, as well as an increase in fruit set percentage [54,55]. Some studies suggest that both the promoter and the phytotoxic effects that AgNPs can generate are related to the Ag ion content and not to the size of the nanoparticles [56]; for example, AgNPs synthesized from extracts of *Moringa oleifera* Lam. (Moringaceae) improved important growth parameters such as germination, root and shoot length, as well as vigor on *Citrus reticulata* Blanco (Rutaceae) seedlings at concentrations of 30µg/mL AgNPs [57].

In addition, nanoparticles show translocation in some plant species, e.g., they have the ability to penetrate the seed coat, store in vacuoles and pass through the plant cell wall. [58]. Therefore, the application of AgNPs for the control of *D. citri* is a potential tool for the management of the insect, as well as in the growth of citrus crop in which they are applied.

## 5. Conclusions

AgNPs showed a high insecticidal effect on second instar nymphs of *D. citri* under laboratory conditions, essentially at concentrations of 32 and 64 ppm, which caused mortalities of 97 and 100%, respectively.

AgNPs showed considerable insecticide effect, with 70 and 80% mortality at 72 h at concentrations of 64 and 128 ppm under greenhouse conditions.

The insecticidal effect shown by AgNPs synthesized from pecan nut shells can be applied as an alternative control agent to be integrated into a psyllid management program; however, field evaluations should be carried out to complement these results.

## Figures and Tables

**Figure 1 insects-15-00469-f001:**
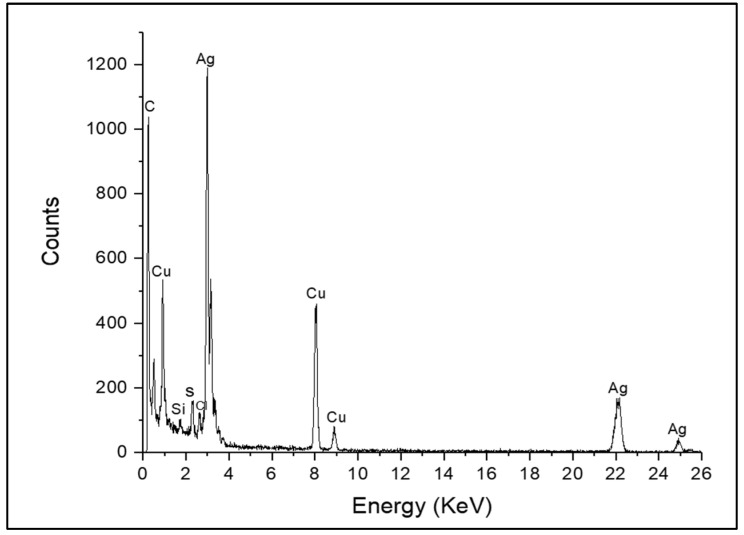
Compositional profile by EDX analysis of AgNPs. The absorption amplicon count shows silver (Ag) as the main component of the nanoparticles and, in minor proportion, elements such as copper (Cu), silicon (Si) and sulfur (S).

**Figure 2 insects-15-00469-f002:**
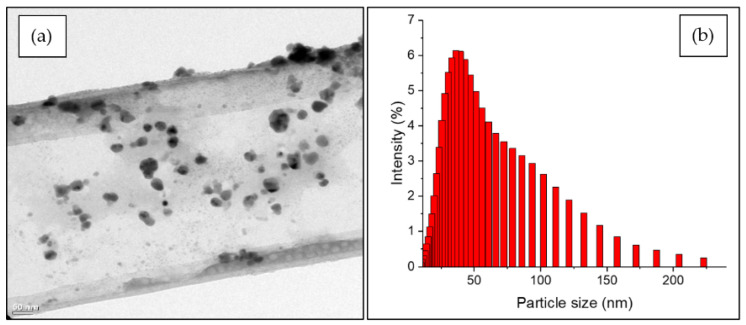
(**a**) TEM micrograph showing the spherical morphology and size of AgNPs. (**b**) Dynamic light scattering (DLS) histogram showing the dispersion of nanoparticle sizes.

**Table 1 insects-15-00469-t001:** Percent mortality and lethal concentrations (LC_50_ and LC_95_) of AgNPs over *D. citri* under laboratory conditions.

% Mortality (mean ± S.E)			
Concentration	24 h	48 h	72 h
0 ppm	2.32 ± 1.23 a	4.28 ± 1.22 a	6.43 ± 1.24 a
2 ppm	21.65 ± 2.62 b	66.70 ± 3.99 b	92.35 ± 1.36 b
4 ppm	24.32 ± 4.05 bc	77.81 ± 4.07 bc	94.58 ± 1.83 bc
8 ppm	34.87 ± 4.56 bcd	82.45 ± 5.35 bc	96.10 ± 1.95 bc
16 ppm	41.01 ± 4.46 cde	86.63 ± 2.21 c	96.66 ± 1.15 bc
32 ppm	45.83 ± 2.26 de	87.40 ± 6.08 c	97.84 ± 0.76 bc
64 ppm	55.21 ± 7.64 e	90.66 ± 2.65 c	100 ± 0.00 c
g.l	6	6	6
F	16.84	57.88	646.94
*Pr* < F	<0.0001 ***	<0.0001 ***	<0.0001 ***
*R* ^2^	0.83	0.95	0.96
^&^LC^50^ (^$^LF 95%)	40.89 (25.84–87.22)	0.23 (0.01–0.74)	0.01 (4.26 × 10^−8^–0.15)
^&^LC^95^ (^$^LF 95%)	18,845 (2926–759,106)	189.25 (70.26–1864)	4.97 (1.47–10.89)

Data transformed by arcsine square root. Means with the same letter are not statistically different (Tukey *p* < 0.05). *** indicate significant contrast to the F value to *p* < 0.001). ^&^ Lethal concentration, ^$^ Fiducial limits. S.E: standard error.

**Table 2 insects-15-00469-t002:** Percent mortality and lethal concentrations (LC^50^ and LC^95^) of AgNPs over *D. citri* under greenhouse conditions.

% Mortality (mean ± S.E)			
Concentration	24 h	48 h	72 h
0 ppm	2.08 ± 2.08 a	3.27 ± 2.18 a	9.01 ± 3.94 a
16 ppm	5.97 ± 1.45 a	8.04 ± 1.23 ab	13.19 ± 1.59 a
32 ppm	10.6 ± 3.34 a	27.75 ± 6.98 b	50.57 ± 9.39 b
64 ppm	55.08 ± 8.69 b	64.19 ± 9.54 c	78.69 ± 6.07 c
128 ppm	56.99 ± 2.74 b	70.14 ± 4.38 c	80.14 ± 4.59 c
g.l	4	4	4
F	34.89	27.64	34.42
*Pr* < F	<0.0001 ***	<0.0001 ***	<0.0001 ***
*Pr* < F *Bloque*	0.07	0.124	0.185
*R* ^2^	0.72	0.78	0.65
^&^LC^50^ (^$^LF 95%)	85.45 (72.87–104.16)	58.90 (51.03–68.86)	37.93 (32.53–43.72)
^&^LC^95^ (^$^LF 95%)	496.53 (331.14–916.35)	334.65 (238.03–550.28)	212.66 (159.10–322.39)

Data transformed by arcsine square root. Means with the same letter are not statistically different (Tukey *p* < 0.05). *** indicate significant contrast to the F value to *p* < 0.001). ^&^ Lethal concentration, ^$^ Fiducial limits. S.E: standard error.

## Data Availability

All the data incorporated in the manuscript.

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
