# Peer review of "Insecticide Efficacy of Green Synthesis Silver Nanoparticles on Diaphorina citri Kuwayama (Hemiptera: Liviidae)"

_insects, 2024, doi:10.3390/insects15070469_

Round 1
Reviewer 1 Report
Comments and Suggestions for Authors The article “Insecticide efficacy of green synthesis silver nanoparticles on Diaphorina citri Kuwayama (Hemiptera: Liviidae)” reports the laboratory and greenhouse investigations of control efficiency of synthesis silver nanoparticles on the Asian citrus psyllid, a vector of Huanglongbing disease (HLB) causal agent bacteria Candidatus asiaticus and Candidatus americanus. The results provide a new promising way in using such nanoparticles to control HLB vector. The article can be published as a brief report after it is revised addressing the following concerns: (1) Abstract: L28-30, check the grammar. This is not a complete sentence. It should be: “Diaphorina citri Kuwayama (Hemiptera: Liviidae) is a vector of ……”. L33, there should be a period (.) between “evaluated” and “Mortality”. Such grammatical errors or word misspellings should be checked and corrected throughout the manuscript. (2) L39-40, key words: “Diaphorina citri” should be added to the key word list, and “nanotechnology” can be removed. (3) MM: L109-110. Why chose 2nd instar nymphs for bioassay experiments? Please explain. (4) Results: L126 (3.4) and L139 (3.5), in italic titles, scientific name “Diaphorina citri” should be in upright letters. (5) Figure 1 and Figure 2, in addition to a brief title, there should be some notes providing further information of the figure, for example, experimental conditions for Figure 1, meanings of each of 2 subfigures for Figure 2, etc. (6) In title of Table 1 and Table 2, “(50 and 95)” should be “LC50” and “LC95” Comments on the Quality of English LanguageEnglish is generally well, with a need for grammatical or spelling checking throughout the manuscript.
Author Response
Response to Reviewer 1
Comments
Thank you for taking the time to review this manuscript. Below find the responses for the comments and they marked with red over the manuscript.
|
Comment |
Response |
|
(1) Abstract: L28-30, check the grammar. This is not a complete sentence. It should be: “Diaphorina citri Kuwayama (Hemiptera: Liviidae) is a vector of ……”. L33, there should be a period (.) between “evaluated” and “Mortality”. Such grammatical errors or word misspellings should be checked and corrected throughout the manuscript. |
The comment is accepted and is marked in red in the text. |
|
(2) L39-40, key words: “Diaphorina citri” should be added to the key word list, and “nanotechnology” can be removed. |
The comment is accepted and is marked in red in the text. |
|
(3) MM: L109-110. Why chose 2nd instar nymphs for bioassay experiments? Please explain. |
The use of 2nd instar nymphs was considered according to IRAC methodology which is suggested for insecticide testing on the insect. it is the testing stage, where control actions begin. |
|
(4) Results: L126 (3.4) and L139 (3.5), in italic titles, scientific name “Diaphorina citri” should be in upright letters. |
They can be improved but the titles are in the style indicated in the instructions for authors of the journal. |
|
(5) Figure 1 and Figure 2, in addition to a brief title, there should be some notes providing further information of the figure, for example, experimental conditions for Figure 1, meanings of each of 2 subfigures for Figure 2, etc. |
The comment is accepted and is marked in red in the text. |
|
(6) In title of Table 1 and Table 2, “(50 and 95)” should be “LC50” and “LC95” |
The comment is accepted and is marked in red in the text. |
(3) MM: L109-110. Why chose 2nd instar nymphs for bioassay experiments? Please explain.
Reviewer 2 Report
Comments and Suggestions for Authors
Dear authors,
thank you for your manuscript. See below some suggestion for its improvement.
introduction
could be nice to introduce citrus..... like which plant families are we talking about here, and perhaps clarify the varieties and hibridisation within the species which led to the different known citrus fruits (just briefly, to give a clear background of the hosts we are looking at). also you can then introduce here the term citriculture ( which comes in l 55).
citrus are in temperate mediterranean regions too. would Italy, Spain, be considered tropical regions?
l61.On the other hand, in addition
Do you have information on the effecs of AgNPs on pollinators? is this something that is investigated?
method
greenhouse conditions: what are there?
I am not sure I understood. Did you apply the insectivide on already infested plants? or on healthy plants then introduced the insects?
results
figure 1. what is EADX?
full names of AgNPs . Iclude also the names of the compounds in full: C, Cu etc. and a brief description of what weare looking at: counts of what, energy of what )
figure 2. same as above, provide a fuller description to understand the figure. can you also label the different parts on the picture? what is the bar graph representing respectingto the picture?
table 1 and 2. same as above, provide full names of abbreviations and species (e.g. what is E.E)
discussion
regarding the potential as a management strategy, what would be the effects or potential effects to look for in pollinators and beneficial insects?
are these likely to be persistent in the environment? soil or get through the plant tissues too?
how cost effective would be this method? thus competititve with the pesticides?
Author Response
Response to Reviewer 2
Comments
Thank you very much for taking the time to review this manuscript. Please find the detailed responses below and the corresponding revisions/corrections highlighted/in track changes in the re-submitted files.
|
Comment |
Response |
|
could be nice to introduce citrus..... like which plant families are we talking about here, and perhaps clarify the varieties and hibridisation within the species which led to the different known citrus fruits (just briefly, to give a clear background of the hosts we are looking at). also you can then introduce here the term citriculture ( which comes in l 55). |
The comment can be implemented. The family to which they belong as well as examples of citrus fruits are written in L56 and marked with yellow. |
|
citrus are in temperate mediterranean regions too. would Italy, Spain, be considered tropical regions? |
The comment is implemented in the manuscript and marked with yellow. |
|
l61.On the other hand, in addition |
The comment was corrected and is marked with yellow in the text. |
|
Do you have information on the effecs of AgNPs on pollinators? is this something that is investigated? |
There is some research on the evaluation of nanoparticles on bees. However, we believe that this information can be taken into account for further work. |
|
greenhouse conditions: what are there? |
Temperature range has been added and marked with yellow (L105). |
|
I am not sure I understood. Did you apply the insectivide on already infested plants? or on healthy plants then introduced the insects? |
Healthy plants with vegetative shoots were infested and then the nanoparticles were applied. It is mentioned in the L141-144 marked with yellow |
|
figure 1. what is EADX? |
The correct name is: EDX: Energy Dispersive X-ray Analysis (L123). |
|
full names of AgNPs . Iclude also the names of the compounds in full: C, Cu etc. and a brief description of what weare looking at: counts of what, energy of what ) |
The elements in Figure. 1 was described and marked with yellow. |
|
figure 2. same as above, provide a fuller description to understand the figure. can you also label the different parts on the picture? what is the bar graph representing respectingto the picture? |
The comment was implemented, both parts of the figure were marked, and a brief description is given. |
|
table 1 and 2. same as above, provide full names of abbreviations and species (e.g. what is E.E) |
E.E is an error of traduction. The correct way is S.E that mean standard error. The abbreviation is found under the table. |
|
regarding the potential as a management strategy, what would be the effects or potential effects to look for in pollinators and beneficial insects? |
We believe that this information can be taken into account for further work. |
|
are these likely to be persistent in the environment? soil or get through the plant tissues too? |
The comment was improved, a brief explanation was added and marked with yellow (L83-84). |
|
how cost effective would be this method? thus competititve with the pesticides? |
There is no exact information about the cost relationship of the application of nanoparticles, because field studies are limited and therefore the cost in comparison with a pesticide is not available. |